# Progress in Plant Nitric Oxide Studies: Implications for Phytopathology and Plant Protection

**DOI:** 10.3390/ijms26052087

**Published:** 2025-02-27

**Authors:** Michaela Sedlářová, Tereza Jedelská, Aleš Lebeda, Marek Petřivalský

**Affiliations:** 1Department of Botany, Faculty of Science, Palacký University Olomouc, Šlechtitelů 27, 779 00 Olomouc-Holice, Czech Republic; ales.lebeda@upol.cz; 2Department of Biochemisty, Faculty of Science, Palacký University Olomouc, Šlechtitelů 27, 779 00 Olomouc-Holice, Czech Republic; tereza.jedelska@upol.cz (T.J.); marek.petrivalsky@upol.cz (M.P.)

**Keywords:** nitric oxide, plant immunity, phytopathogens, stress signaling, nanomaterials

## Abstract

Nitric oxide (NO) is a gaseous free radical known to modulate plant metabolism through crosstalk with phytohormones (especially ABA, SA, JA, and ethylene) and other signaling molecules (ROS, H_2_S, melatonin), and to regulate gene expression (by influencing DNA methylation and histone acetylation) as well as protein function through post-translational modifications (cysteine S-nitrosation, metal nitrosation, tyrosine nitration, nitroalkylation). Recently, NO has gained attention as a molecule promoting crop resistance to stress conditions. Herein, we review innovations from the NO field and nanotechnology on an up-to-date phytopathological background.

## 1. Introduction

Plants are exposed to various stress factors during their lifetime and have developed efficient defense mechanisms necessary for survival in their natural environment. The plant defense system comprises two basic realms, i.e., constitutive defense components and induced defense responses. The second one depends on multiple interconnected cellular and physiological reactions with the involvement and participation of various molecules or effectors [1]. Reactive oxygen species (ROS) and reactive nitrogen species (RNS) are intracellular signaling molecules or effectors of the defense responses. In plant biology, recent studies especially pointed at nitric oxide (NO) as a redox-active molecule [2,3]. The last three decades of nitric oxide research have elucidated the biosynthetic pathways and multiple roles of this molecule in plant physiology (germination, growth, flowering, senescence) and pathophysiology (coping with abiotic and biotic stresses) [4,5,6]. For plants attacked by pests or infested by pathogens, strong evidence was gathered for NO crosstalk with the phytohormones salicylic acid (SA), jasmonic acid (JA), and ethylene; less is known of biotic stress-induced NO interference with abscisic acid (ABA), auxins, brassinosteroids, cytokinins, gibberelins, melatonin, or polyamine metabolism, which relate to ontogeny and abiotic stress responses [7,8,9,10,11]. Intra- and intercellular signaling were newly complemented with hints that NO/RNS and derived molecules, e.g., S-nitrosoglutathione (GSNO) and nitrated fatty acids (NO_2_-FAs), also transmit signals within and among plant organs as well as between individuals in their vicinity [3], which raises questions of knowledge applicability. Nanotechnologies progress, prompted by a demand to increase crop resilience to climate change and to improve plant protection against pests and pathogens, and the production of micro-/nanoparticles enables plant/organ-targeted delivery of desired chemicals, including NO donors [12]. Herein, we aim to gather and discuss novelties in NO research and plant–pathogen interactions, from basic studies to their possible applications.

## 2. NO Synthesis and Degradation in Plants

NO occurs as a gaseous free radical with one unpaired electron. Removing or adding an electron provides chemically related reactive nitrogen species, nitrosonium cation (NO^+^) or nitroxyl anion (NO^−^), respectively, but with distinct biological properties [13]. Due to its small size and lipophilic character, NO can diffuse freely through cell membranes, and the half-life of NO in the cell is around 3–5 s, which is crucial for its biological activity [14]. NO reacts with the highly reactive superoxide anion radical (O_2_^−^) to form peroxynitrite (ONOO^−^), a potent nitrating compound. NO and RNS can react with free thiol groups to form S-nitrosothiols (SNOs) (Figure 1). NO can also nitrate proteins (Figure 1), unsaturated fatty acids in lipids, and nucleic acids [15].

The molecular mechanisms of NO biosynthesis in plants under different conditions are still subject to debate. NO can originate from different routes and substrates in plants. Many studies confirm NO formation by enzymatic and non-enzymatic mechanisms, depending on the organism, the location, and the factors that stimulate its formation [16]. Figure 1 summarizes the pathways of NO production and conversion in plants. The best-described and -evidenced pathway of NO synthesis in plants is reductive, where NO is generated from nitrate (NO_3_^−^) (Figure 1). Nitrate reductase (NR, EC 1.7.1.1), located in the cytosol, represents a well-studied enzyme shown to produce NO under in vitro and in vivo conditions. NR catalyzes the NAD(P)H-dependent two-electron reduction of NO_3_^−^ to nitrite (NO_2_^−^) under conditions of a higher NO_3_^−^ concentration and a lower oxygen concentration in the cell. Depending on the plant source, the molecular weight of the NR homodimer ranges from 200 to 250 kDa. Each monomer contains three prosthetic groups: FAD, heme, and a molybdenum cofactor. NR activity is post-translationally regulated by reversible phosphorylation. NR is one of the key enzymes in NO biosynthesis in roots [17], and following these findings, further studies have confirmed that it is the first uniquely identified source of NO in plants [18,19,20,21,22]. A specific NO-producing enzyme, nitrite: NO reductase (NiNOR, EC 1.7.2.1), has been localized in the plasma membrane of tobacco root cells (Figure 1). The electron donor for NiNOR-mediated NO_2_^−^ reduction is cytochrome c, not NAD(P)H [23]. In collaboration with the molybdenum-containing amidoxime reductant (ARC), NR can produce NO from NO_2_^−^ in *Chlamydomonas*, even at high NO_3_^−^ concentrations [24]. In *Arabidopsis*, ARC proteins are not involved in NO_2_^−^-dependent NO production [25].

Several other pathways that lead to NO production have been uncovered (Figure 1). Under hypoxic conditions, NO production is catalyzed by animal xanthine oxidoreductase (XOR, EC 1.17.3.2) [26], and in plants, the activity of XOR was detected in pea peroxisomes [27]. A recent study [28] shed light on NO production in mitochondrial cytochrome c oxidase (COX) and suggested the involvement of its subunits in NO_2_^−^-dependent NO production (Figure 1).

Non-enzymatic sources also contribute to NO production under specific conditions (Figure 1). In the apoplast, where the pH is acidic, NO_2_^−^ dismutates to NO_3_^−^ and NO [29]. NO_2_^−^ can also be reduced to NO and dehydroascorbic acid in the aleurone layer of barley [30]. In chloroplast membranes, the conversion of NO_2_^−^ to NO under light conditions is catalyzed by carotenoids in vitro [31].

Multiple studies have tried to demonstrate the existence of an oxidative route for NO production in plants. The arginine-dependent NO production mediated by the enzyme NO synthase (NOS, EC 1.14.13.39) (Figure 1) is very well described in animals, bacteria, and fungi. NOS catalyzes the oxidation of the guanidine nitrogen of L-arginine to release NO and form L-citrulline. The reaction proceeds through double mono-oxygenation, using molecular O_2_ and NADPH as co-substrates and FAD, FMN, and tetrahydrobiopterin as cofactors. In plants, there are many results indicating the presence of NOS activity, but the relevant proteins and genes are not identified. An unsuccessful search for transcripts encoding NOS-like proteins in a dataset of 1000 plant genomes [32] argues against NOS in plants. No typical sequences homologous to mammalian NOS have been found in plants in those species in which previous studies reported NOS activity, nor has any effect of animal NOS inhibitors been recorded. Key insights into the issue of plant NOS were provided by the study of Foresi et al. (2010) [33], where the NOS from the green alga *Ostreococcus tauri* (OtNOS) was comprehensively characterized. Later, the first NOS in cyanobacteria was characterized in *Synechococcus* PCC 7335 [34]. Nevertheless, such evidence of a functional NOS enzyme is still lacking in higher plants.

Other oxidative production routes of NO in plants have been described from hydroxylamine (Figure 1) or salicyl hydroxamate [35]. Enzymes of polyamine catabolism, such as copper-diamine oxidase (EC 1.4.3.22) and polyamine oxidase (EC 1.5.3.13), are reported to be indirectly involved in NO production via an unknown mechanism (Figure 1) [36,37]. Recently, a new oxidative pathway of NO production from oximes was described in plants (Figure 1). Oximes, such as indole-3-acetaldoxime (precursor to auxine indole-3-acetic acid), are intermediate oxidation products in NO synthesis catalyzed by peroxidase (EC 1.11.1.7) [38].

## 3. NO-Mediated Post-Translational Protein Modifications

Important parts of NO signaling pathways in plants are mediated by post-translational modifications (PTMs) of proteins, executed by NO, RNS, or NO_2_-FAs. S-nitrosation and tyrosine nitration are considered the most relevant in transducing bioactivity during stress responses [39,40,41].

The nitration of proteins consists of the insertion of a nitro group (-NO_2_) in the ortho-position relative to the hydroxyl of the benzene core of tyrosine (leading to 3-nitro-tyrosine) or tryptophane (4-nitro-tryptophan or 6-nitro tryptophan) (Figure 1), thus irreversibly changing the conformation of the modified protein and consequently affecting its biological activity [42]. It is reported that only 1 to 5 Tyr residues out of 10,000 are modified under physiological conditions [43]. Three critical factors influence the selectivity of this PTM: (1) the availability of nitrating agents, (2) the availability of the protein and accessibility of Tyr/Trp residues, and (3) the primary sequence surrounding the potentially nitrated Tyr/Trp residue [44]. Electrophilic substitution of tyrosine with the -NO_2_ group causes a decrease in the pKa of the hydroxyl group, which may subsequently reduce its reactivity for phosphorylation. The nitration and phosphorylation of Tyr residues can be competitive processes, the extent of which will be influenced by local RNS levels [42]. The regulation of several metabolic pathways by protein nitration has been described in plants under physiological and stress conditions. It was shown that nitration regulates the activity of antioxidant enzymes involved in the scavenging of ROS in cells [41]. Studies of cytosolic pea ascorbate peroxidase (PsAPX, EC 1.11.1.11) have provided the first evidence of structural and functional protein changes due to nitration [45,46]. Tyr235 was identified by proteomic approaches and in silico as the most likely target of nitration due to its localization at a distance of 3.6 Å from the prosthetic heme group at the bottom of the catalytic pocket. The nitration of Tyr235 significantly reduces the activity of PsAPX, presumably by disrupting its structure [45]. Monodehydroascorbate reductase (MDAR, EC 1.6.5.4), crucial for ascorbate regeneration, is another important point in the ascorbate–glutathione cycle which is regulated by nitration. In recombinant pea MDAR, Tyr213, Tyr292, and Tyr345 are nitrated, and site-directed mutagenesis has confirmed that Tyr345 is the key residue whose nitration activity in MDAR is reduced. Tyr345 is located at a distance of 3.3 Å from His313, which is an important component of the site for NADP^+^ cofactor binding, and thus the nitration of Tyr345 affects proper cofactor binding [46]. Superoxide dismutases (SODs, EC 1.15.1.1) are a group of metalloenzymes that catalyze the disproportionation of O_2_^−^ to hydrogen peroxide. Nitration inhibited the activities of Mn-SOD1, Fe-SOD3, and CuZnSOD3, although with different intensities [47].

S-nitrosation, the reversible modification of cysteine residues in proteins (Figure 1), represents one of the key NO-mediated redox signaling pathways [48]. S-nitrosothiols (SNOs) are formed by the covalent binding of a nitroso (NO-) group to the sulfhydryl (SH) group of the target cysteine residue. S-nitrosation changes the structure and function of many proteins, e.g., enzyme activity, subcellular localization, or changes in interactions with binding partners [49]. Proteomic approaches have revealed several S-nitrosation-modified proteins that regulate physiological and pathophysiological processes in plants. Recent findings point to a key role of S-nitrosation in the biosynthesis of plant hormones, programmed cell death, and regulation of transcription through the S-nitrosation of nuclear proteins [50,51]. S-nitrosation is also an essential regulator of the activity of antioxidant enzymes of the ascorbate–glutathione cycle and is involved in the regulation of the SA signaling pathway, which is critical in plant immune responses or in the hypersensitive response (HR) [40,52,53].

GSNO, the most abundant low-molecular-weight SNO, serves as the primary storage and transport form of NO in cells (Figure 1) [54]. GSNO mediates transnitrosation reactions, whereby the NO group is transferred to the thiol group of another cysteine to form a new SNO [55]. In plants, two key enzymes involved in SNO degradation have been described: S-nitrosoglutathione reductase (GSNOR, EC 1.1.1.284) and thioredoxin reductase (TRXR, EC 1.8.1.9). The balance between low-molecular-weight SNOs and S-nitrosated proteins is indirectly controlled by the GSNOR-mediated denitrosation of GSNOs (Figure 1) [56,57]. The second mechanism involves the thioredoxin system (TRXR-TRXh5), consisting of TRXR, thioredoxin h5 (TRXh5), and NADPH (Figure 1). In contrast to the indirect control of S-nitrosated protein metabolism in plants controlled by GSNOR, TRXR-TRXh5 represents an entirely different pathway of protein denitrosation—converting other protein SNOs and also selectively discriminating its substrates during plant immune reactions [58,59,60].

Proteomic studies in *Arabidopsis* mutants have revealed the specific role of the aldo-keto reductase family (AKR) in the NADPH-dependent regulation of protein SNOs and NO homeostasis in plant cells (Figure 1) [61]. Human AKR (AKR1A1) has been described as being involved in GSNO catabolism, similar to GSNOR [62]. AKRs are monomeric NADPH-dependent oxidoreductases sharing a common structural motif, a conserved cofactor binding domain, and a conserved catalytic tetrad [63]. AKRs are relatively widespread because their substrates are diverse reactive carbonyl compounds, such as ketones and aldehydes, which are reduced to the corresponding alcohols. AKRs detoxify compounds produced during stress conditions, and their expression is induced by various biotic or abiotic stresses [64,65,66,67,68,69]. The GSNOR enzyme uses NADH as the reducing equivalent, whereas AKR is strictly NADPH-dependent [70]. Thus, in addition to the activity of GSNOR, the enzymatic reduction of GSNO to glutathione sulfinamide catalyzed by AKR may participate in NO homeostasis and control NO-related biological functions under physiological and stress conditions. The current studies shed light on the vital role of AKR in NO/GSNO homeostasis in plants [60,61,71].

Increased production of O_2_^−^, ONOO^−^, and GSNO (Figure 1) can lead to structural and, consequently, functional changes in several macromolecules, causing the development of oxidative stress (lipid peroxidation, protein carbonylation) and nitrosative stress (lipid and protein nitration, S-nitrosation) [72]. Nitrosative stress also causes a disturbance in the balance of nitrosation between low-molecular-weight and protein SNOs. Under certain stress conditions, these processes co-occur and are called nitro-oxidative stress in the context of plant biology [73]. In plants, an essential role in defense against nitrosative stress is played by GSNO and GSNOR, which catalyze the NADH-dependent reduction of GSNO to produce oxidized glutathione (GSSG) and ammonia (Figure 1) [52]. GSNOR is a major contributor to intracellular NO metabolism due to its ability to metabolize GSNO and is indirectly involved in the regulation of S-nitrosated protein levels through transnitrosation reactions, where GSNO reacts with protein thiols to form S-nitrosated proteins [56]. GSNO, a stable storage and transport form of NO in vivo, is generated by the S-nitrosation of reduced glutathione (GSH) occurring indirectly through the formation of N_2_O_3_ or by a direct reaction of NO with the glutathione thiyl radical as a reaction intermediate [74]. The GSH/GSNO ratio could indicate the cell’s redox state [75,76].

## 4. Lipid-Mediated NO Signaling

Polyunsaturated fatty acids are known to react with NO and derivative forms to form nitrated fatty acids (NO_2_-FAs). NO_2_-FAs serve as NO-releasing signaling molecules, but can also reversibly esterify with complex lipids or modulate protein function through a PTM called nitroalkylation [77,78]. Available studies have shown that NO_2_-FAs are involved in signaling during plant development and participate in defense responses against abiotic stress conditions [79,80,81,82]. Aranda-Caño et al. [83] identified previously unknown NO_2_-FA storage biomolecules in *Arabidopsis* and described their distribution during plant development. The esterified NO_2_-FAs were detected as nitro-linolenic acid (NO_2_-Ln) and, for the first time in *Arabidopsis*, nitro-oleic acid and nitro-linoleic acid. These results indicate the importance of NO_2_-FA esterification in phospholipids and proteins, particularly in its involvement in biomembrane dynamics and signaling processes during plant development, and they open up a new research field to study. Efforts have been made to outline general models for the roles of NO_2_-FAs in both abiotic stress and plant defense [5].

## 5. Plant Immunity Concepts

Evidence growing over time indicates that NO influences plant interactions with associated organisms (viruses, bacteria, oomycetes, fungi, protozoa, animals, plants) and relates to symbiotic interactions (from mutualism to parasitism) and herbivore attack [9,84,85,86,87]. Bacteria, oomycetes, and fungi produce NO during plant pathogenesis (Figure 2A). To alleviate the stress impact, plants employ structural and biochemical defense mechanisms, both preformed and inducible. Generally, plants prioritize preformed first-line defense strategies in the apoplast (redox changes, defensins, small peptides, secondary metabolites) over investment in induced symplastic processes [88]. The interplay between host and microbe genetics, environmental conditions, and external stimuli (other associated organisms, signals, pesticides) decides between health and disease.

Plant immunity combines two mutually interacting components at the molecular level (Figure 2B): (1) pattern-triggered immunity (PTI) based on cell surface pattern recognition receptors (PRRs) with or without a kinase domain, and (2) effector-triggered immunity (ETI) employing intracellular nucleotide-binding leucine-rich repeat receptors (NBS-LRRs or NLRs) to detect effectors secreted by pathogens (both included in the Zig-Zag model of plant–pathogen coevolution) [89]. The activation of PTI was first reported to precede ETI, but both are interlinked more than previously believed. PTI has been interpreted as a conserved defense mechanism in which PRRs bind molecules called “molecular patterns”, i.e., “pathogen-, microbe-, damage-, and herbivore-associated molecular patterns” (PAMPs, MAMPs, DAMPs, and HAMPs) [90,91]. ETI represents defense induced by an effector, a specialized molecule produced by pathogens to overcome PTI, binding to NLRs (members of a large family of signal transduction ATPases with numerous domains, kept in a “pre-activation” state through inter-domain interactions) in the plant cytoplasm, nucleus, plasma membrane, and tonoplast [92,93,94]. Recent experiments in *A. thaliana* showed that signaling initiated both by PRRs and NLRs induces largely overlapping downstream cellular responses and that only mutual PTI and ETI orchestration leads to plant cell resistance [95,96]. ETI is predominantly activated in conjunction with PTI, while ETI activation without PTI is relatively rare [97]. Plant defense is a multi-stage process, influenced by the cell type, organ (the original paradigm arose from studies on leaves; at present, roots are studied in detail) and its ontogenetic stage, and the interacting organisms, and interpretation differs by author [88,94,98,99]. In general, the series of processes includes Ca^2+^ influx, ROS/RNS/H_2_S production, redox changes, increased cADPR and cGMP levels, activation of the MAP kinase cascade and transcription factors, formation of PR proteins, protein PTMs, activation of signaling pathways (SA in biotrophs vs. JA and ethylene in necrotrophs), cytoskeleton reorganization, HR, callose deposition, and reinforcement of the cell wall [93,100].

The plant recognition and mechanisms leading to immunity might be perceived from different perspectives; thus, changes in the categorization and terminology of plant immunity are currently being discussed [95,101,102], with a preference for molecules that trigger a response (PTI vs. ETI), the types of receptors (PRR- vs. NLR-mediated immunity), localization in which a response is activated (extracellularly vs. intracellularly triggered immunity), or the location of immune receptors (surface-receptor-triggered vs. intracellular-receptor-triggered immunity). However, the mechanism of how a plant differentiates between beneficial and harmful microorganisms still remains unexplained [98,99]. Distinguishing potential pathogens from mutualists prior to energy investment in immune or symbiotic responses has been proposed by dual or multiple recognition, e.g., coupled pattern sensing and combining patterns with effector molecules [103]. Since NO was first linked with plant resistance to bacterial invasion [84], it has been found to function in various plant–microbe associations [104], underscoring redox changes and signaling cascades at the local level (both in PTI and ETI, although NO can mediate downstream immune outputs differing among tissues and organs) [86], and also probably at the systemic [105,106] and interplant levels of communication [3].

## 6. A Piece of NO in the Plant Immunity Puzzle

The role of NO in plant disease resistance has been studied for almost three decades [84]. Attention has primarily been focused on HR, the main defense mechanism of plants against attack by biotrophs and hemibiotrophs. NO was originally connected with HR [84], and this obsolete information unfortunately still persists in some reviews (e.g., [107]); nonetheless, the HR molecular mechanism can be explained as follows: NLRs modify their conformation upon binding effectors from (hemi)biotrophic pathogens to arrange a pentameric wheel-like complex newly termed a resistosome, which acts as calcium channel [108], or their N-terminal α helices forming a funnel-shaped structure may perturb the plasma membrane integrity and induce cell death [109]. HR thus represents suicide of the individual infected cell or of a few cells surrounding the zone of pathogen ingress and contributes both to local ETI and systemic defense signaling. Therefore, an increased NO level is not the primary cause of HR.

Nevertheless, NO orchestrates other defense mechanisms (Figure 2B). Plant immunity against pathogen infection (namely bacteria) has been inseparably associated with S-nitrosation as the principal signaling route. This also influences the regulation of ROS production, primarily generated by plant NADPH oxidases (also termed respiratory burst oxidase homologs, RBOHs) which can be targeted by NO- and H_2_S-dependent PTMs [110]. Many works on plants like tobacco and *A. thaliana* indicated a close interrelationship between RNS and ROS metabolism. Lee et al. [111] showed that RBOHD is down-regulated by C-terminal phosphorylation and ubiquitination while being activated by its N-terminal phosphorylation following PAMP recognition (Figure 2B). More recently, a molecular mechanism of NO involvement in ROS burst regulation was explained in the *A. thaliana–Pseudomonas syringae* pv. *tomato* interaction [112]. The recognition of PAMPs like flagellin or elongation factor Tu by respective membrane-located PRRs leads to a rapid nitrosative burst; the accumulation of NO subsequently leads to the S-nitrosation of the receptor-like cytoplasmic kinase (RLCK) botrytis-induced kinase 1 (BIK1) at Cys80, promoting the phosphorylation of BIK1, thereby enhancing both the stability of BIK1 and its interaction with RBOHD, which promotes ROS accumulation in the apoplast. Moreover, it influences the BIK1-mediated phosphorylation of Ca^2+^-permeable channels, leading to calcium influx and stomatal closure [113,114]. Concurrently, callose deposition (which might indicate interference with ABA signaling) and PTI-related gene expression are hindered [112]. When NO accumulates at later stages of PTI, it may S-nitrosate AtRBOHD at Cys890, decreasing ROS synthesis, similar to what happens during ETI [115], which further supports the interconnection of PTI/ETI outlined in Figure 2B [90,93]. Fine spatiotemporal tuning of (S)NO concentrations is essential in order to retain or suppress the individual defense mechanisms, executed at the local scale indirectly by GSNOR [56] or selectively by TRX [58].

NO was also reported to stimulate the antioxidant system; therefore, a positive effect of NO in low doses might be attributed to optimizing the cell redox state [116]. The reaction of NO with O_2_^−^ leads to the formation of highly reactive ONOO^−^ reported to induce protein nitration and accumulation of SA-responsive pathogenesis-related 1 protein (PR1) employed in plant defense responses [72]. Surprisingly, PR1-like proteins from the same evolutionarily conserved CAP superfamily exert an offensive function in both hemibiotrophic and necrotrophic plant pathogenic fungi (*Cytospora* spp., *Fusarium* spp., *Moniliophthora perniciosa*) as virulence factors, and some function as effectors to suppress PTI [117]. Nitrosative stress has lately been discussed as an epigenetic regulator of gene expression, and it seems to be a universal mechanism in many organisms affecting histone deacetylases (HDAs) and thus increasing histone acetylation and gene transcription [50]. Apart from animal models, NO/RNS was proposed to exert this regulation also on stress-induced plant genes [118] and pathogenicity genes in oomycete *Phytophthora infestans* [119], though the molecular mechanism remains elusive.

Multiple studies have delineated that NO regulates plant ontogenetic and stress processes through a complex network of second messengers such as Ca^2+^, cADPR [120], cGMP [121], and lipids [83], but the data are scattered, and many details of signaling during plant–pathogen interactions are still unresolved. In animal cells, the canonical NO signaling pathway includes NOS-produced NO, which activates soluble guanylate cyclase (GC) by binding to iron in its prosthetic heme, leading to increased levels of the second messenger cyclic 3′,5′-guanosine monophosphate (cGMP) [122,123]. In plants, the importance of the NO-cGMP-dependent pathway was delineated, and novel enzymes possessing guanylate cyclase activity were identified, but the function of NO-sensitive GC *in planta* still remains unconfirmed [124,125]. GC activity has been detected in vitro for six receptor proteins from *A. thaliana*. It turns out that one of these six receptors is the NO-dependent GC enzyme (NOGC1) [121]. On the other hand, a comprehensive bioinformatics study conducted on more than 1000 plant species did not confirm the existence of a cGMP-NO signaling pathway in plants [32]. Thus, NO bioactivity in plants is currently believed to be mediated in major part through redox-based modifications of target proteins, such as heme S-nitrosylation or cysteine S-nitrosation [16,52].

At the same time, reorganization of the cytoskeleton within plant host cells during interactions with microorganisms has been reported [126,127,128,129]. Several questions thus await to be answered to unveil NO’s and cGMP’s roles during pathogenesis: (1) Is plant cGMP-dependent phosphorylase involved in tubulin phosphorylation like it is in animal cells? (2) How is tubulin tyrosine nitration involved [130]? (3) Does actin undergo tyrosine nitration [131]? Research of cytoskeletal actin recently brought a novel perspective on plant immunity. In contrast to previous reports, its depolymerization was not linked with higher susceptibility but with higher resistance to pathogens of Brassicaceae (*A. thaliana, Brassica napus*) based on SA pathway activation [132,133].

## 7. Exogenous Application of Nitric Oxide Donors

Endogenous NO levels might be modified by the application of exogenous NO donors. With a short half-life under aerobic conditions, NO, as a lipophilic radical, easily interacts with and diffuses through the plasma membrane. Like other phytohormones, the fine-tuning of NO homeostasis is required [85]. In the early studies of exogenous NO application (NO donors used in laboratory experiments include SNOs like GSNO or SNAP, sodium nitroprusside, and NONOates), a bi-phasic dose/concentration effect called hormesis was reported [134]. Increased NO levels within a “physiological window” (named the “nitrosative door” in seed germination [135]) lead to better plant acclimation and improved resistance, whereas an excessive increase leads to metabolism disturbances. At low concentrations, NO can increase immunity, while at extreme concentrations, it reduces it within minutes due to nitrosative stress, redox changes, and PTMs [12]. At high concentrations of applied NO donors, the effects of NO-related oxidation products, such as NO_2_, must also be considered [136,137]. Thus, in recent years, new materials have been investigated with an impact on the gradual and targeted release of NO from a selected donor molecule to avoid toxicity and to benefit from lower and prolonged NO doses [138]. These are more easily attained by the use of nano NO donors, which seem to be optimal for larger-scale ex planta application in the rhizosphere [139,140], phyllosphere [141], or fruits [142,143].

Following decades of medicinal therapeutic applications of NO linked to its vasodilative, tumoricidal, and antimicrobial effects, the progress in the development of nanomaterials has brought the possibility to deliver NO successively to desirable tissues and cells [137]. Tan and He [138] reviewed a plethora of nanovehicles in which supporting nanomaterials (semiconductor quantum dots, silica nanoparticles, up-conversion nanomaterials, carbon/graphene nanodots, gold nanoparticles, iron oxide nanoparticles) are conjugated with NO donors, including N-diazeniumdiolates, SNOs, nitrosyl metal complexes, and organic nitrates. This represents an inspiration for biotargeting, i.e., precise exogenous NO applications in plant science, from mechanistic studies on plant models in laboratories to possible future use in agriculture as nanonutrients/nanopesticides. Plant defense mechanisms (structural or biochemical) are induced differentially at the cell/tissue/organ/individual/interplant level [88]. Presently, modifications of NP surfaces (coating with peptides, aptamers, signaling tags, and sugar adducts) pledge to enable specific cargo delivery due to specific interactions with the surfaces and biomolecules of plants or their microbial partners [144]. Nevertheless, prior to the practical application of these novel materials, many aspects must be thoroughly taken into account [12,145].

## 8. NO-Based NPs in Crop Protection in Agriculture

The worldwide pressure to increase food production raises demands for more efficient crop cultivation. However, intensive agriculture with genetically homogenic plants enables a burst of epidemic diseases and the fast spread of pests. Strategies of integrated pest management compromise stand and environmental conditions, interacting organisms (to support beneficial microorganisms and the control of pathogens and pests), agriculture practices, and other socioeconomic factors. More “eco-friendly” plant production strategies are on the go, aiming to decrease the amount of chemicals applied to the field [146]. These involve more efficient pesticide application with timing based on predictive models (based on weather forecasting vs. knowledge of individual pathogen/pest life cycles) and successive signaling to farmers, or automated field monitoring systems/application (e.g., remote field sensors, mobile platforms, or drones). To diminish or ideally to replace chemical pesticides, the construction of micro- and nanoparticles with different origins and biological effects has been initiated; application methods for plants such as fumigation, spraying, trunk injection, soil drenching, seed treatment, and baits are chosen in regard to pest biology and transmission [147].

Agriculture might possibly also benefit from newly developed NO-releasing nanomaterials, mainly those that are biodegradable (e.g., based on chitosan nanoparticles). Nanoencapsulated NO donors that provide sustained and localized NO release promise the amelioration of crop growth and yield under stress conditions, as well as during post-harvest fruit storage. However, basic and applied research is needed to understand their mode of action, dosing for specific plant species and genotypes in given environmental conditions, and impact on signaling both within and between plants. Safe technology transfer to plant protection practice should also be preceded by studies which answer the questions of how we influence the whole ecosystem on a long-term scale.

Specifically in the area of crop protection, external application of NO can manipulate endogenous NO levels and the outcome of plant–pathogen interactions (decrease disease symptoms, hold back pathogen reproduction, or kill the pathogen). However, NO sources from both partners must be considered in plant–pathogen interactions (Figure 2A). NO is produced by fungal and oomycete pathogens during their growth, penetration, host colonization, and/or mycotoxin production [97,119,148,149,150,151]. The most promising for plant protection seems to be the application of chitosan-based NPs. In relation to plant immunity, chitosan alone has been tested to stimulate plant defense mechanisms, including NO production; recently, this was shown, e.g., for the *Capsicum annuum*–*Alternaria alternata* interaction (leaf spot disease of chilli) [152]. Improved bioavailability of NO in plant tissues is desired, but as argued by Kandhol et al. [137], during the application of NPs, ROS and NO homeostasis are simultaneously affected. In plant–pathogen studies, this must be thoroughly taken into account to avoid data misinterpretation. Moreover, studies should take into account the effects on beneficial microorganisms in the plant microbiome. Hence, NO-based NPs have been utilized mainly in post-harvest applications so far. NO inhibits ethylene, delays senescence, enhances fruit quality, and inhibits pathogens/saprotrophs. Accordingly, it was exploited to prolong post-harvest storage by NO fumigation and the application of classical NO donors [153] or encapsulated NO donors [142,143]. Manipulation of the plant redox state by NO-based NPs [137] interferes with the growth of aerobic microorganisms (mainly fungi and bacteria), decomposing fruits, vegetables, grains, and other plant commodities.

## 9. Conclusions and Future Directions

This review summarizes the current state of knowledge on NO production in plants and the mechanisms of NO’s functions focusing on plant diseases, including NO’s involvement in mechanisms of pathogen recognition and plant immunity. Major advances have been achieved in the understanding of how NO is produced and degraded in plants; however, there are still important gaps in our detailed knowledge of species-, tissue-, and cell compartment-specific pathways. Nevertheless, NO is undoubtedly one of the key molecules in plant stress signaling, which is mediated by NO-based redox changes in target biomolecules, regulation of gene expression, and post-translational protein modifications, including pathogenesis. The molecular mechanisms of NO’s action are best known from the *Arabidopsis* model and are known to differ in other plant species and agriculturally relevant crops. Molecular data provide the concepts of plant immunity and comprehension of defense mechanisms and highlight the complexity of plant microbiomes. In this field, more studies are needed to understand the molecular mechanisms of NO’s role in the epigenetic regulation of PR genes, the function of NO-regulated GC *in planta*, and NO’s control of cytoskeletal reorganization. Within plant–microbe interactions, lipid-mediated NO signaling also needs further detailed investigations.

Current advances in experimental tools and research methodologies available in plant science are rapidly advancing. The use of CRISPR/CAS editing of genes of components of plant immunity represents a promising strategy to increase crop resistance to both abiotic and biotic stresses. Genetically encoded biosensors for NO, ROS, and other redox molecules provide highly effective and sensitive tools for advanced imaging and monitoring of NO dynamics in plant cells. The increasing sensitivity and accuracy of proteomics analyses enable studies of NO-dependent modulations of redox proteomes at subcellular resolution. Similarly, recent years have witnessed the fast development of prediction tools for the modeling of protein structures influenced by PTM redox modifications, using tools based on machine learning and AI.

Still, studying complex interactions with multiple partners, reflecting how plants interact in nature, remains challenging. We must consider not only the pathogen’s biology and trophic strategy (biotrophy, hemibiotrophy, necrotrophy, sapronectrophy), the site of plant entry, and the affected organs (root, stem, leaf, flower, fruit) and tissues (dermal, ground, vascular, meristems), but also other interacting microbes, plants, animals, and abiotic factors. Nanosized NO donors have a high potential to replace “classical” NO donors both in basic research and applications, with the perspective to be used especially in greenhouse and hydroponic vertical growth containerized systems rather than for field crops, as well as for improved storage of plant products. NO-based NPs have not been used for crop protection against pathogenesis, yet they might be advanced for specific pathogen biotargeting. NO is produced by plants as well as associated microbes and other organisms; thus, modifications of NO by external donors influence the whole plant’s microbiome and plant community. Therefore, future comprehensive research is needed to determine the doses and targets for NO release to trigger effective plant defense signaling under variable environmental conditions.

## Figures and Tables

**Figure 1 ijms-26-02087-f001:**
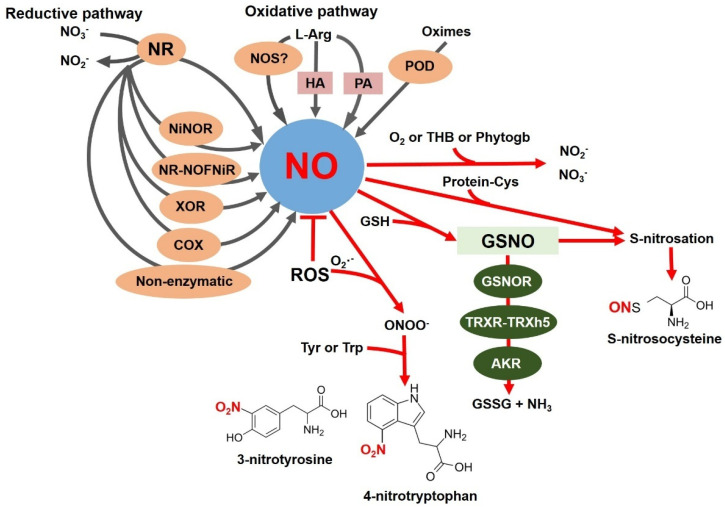
Overview of NO production (black arrows) and conversion (red arrows) pathways in plants. Enzymatic and non-enzymatic reductive pathways using nitrate and nitrite reduction (left): NO is produced by the cytosolic nitrate reductase (NR), nitrite: NO reductase (NiNOR), nitrate reductase-nitric oxide forming nitrite reductase (NR-NOFNiR), xanthine oxidoreductase (XOR), and mitochondrial cytochrome c oxidase (COX). Oxidative pathways of NO synthesis include an NOS-like enzyme using L-Arg as a substrate, undescribed metabolism of hydroxylamine (HA) and polyamines (PAs), and the production of NO from oximes catalyzed by peroxidase (POD). Pathways of NO conversion (right): The reaction of NO with molecular oxygen (O_2_) leads to the formation of nitrate (NO_3_^−^) and nitrite (NO_2_^−^). Phytoglobins (Phytogb) can act as NO dioxygenases and metabolize NO to NO_3_^−^. Truncated hemoglobin (THB) modulates NO levels and NR activity. The transfer of the NO^+^ group to the cysteine residue of reduced glutathione (GSH) forms stable S-nitrosoglutathione (GSNO). The interaction of reactive nitrogen species with free sulfhydryl groups of protein Cys results in S-nitrosation. The enzyme S-nitrosoglutathione reductase (GSNOR) breaks down GSNO to form oxidized glutathione (GSSG) and ammonia (NH_3_). GSNO can be cleaved by the thioredoxin system consisting of thioredoxin reductase (TRXR) and thioredoxin h5 (TRXh5). Aldo-keto reductases (AKRs) form a new class of enzymes involved in NO homeostasis. During prolonged immune activation, GSNOR is regulated through reactive oxygen species (ROS) oxidation. NO reacts with the superoxide anion radical (O_2_^−^) to form peroxynitrite (ONOO^−^), which can cause nitration of tyrosine residues in proteins.

**Figure 2 ijms-26-02087-f002:**
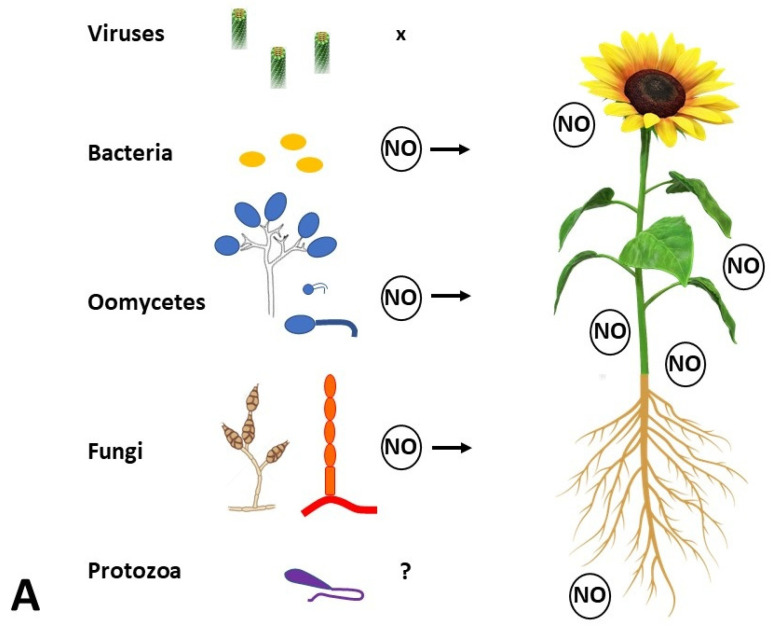
Nitric oxide (NO) is involved in plant interactions with microbes. (**A**) During pathogenesis, NO is produced by the infected plant as well as by bacteria, oomycetes, and fungi themselves. NO production by phytopathogenic protozoa has not been reported. (**B**) Pattern-triggered immunity (PTI), based on molecular pattern recognition receptors (PRRs) at the plant cell surface, is co-activated with effector-triggered immunity (ETI) involving nucleotide-binding leucine-rich repeat receptors (NLRs). The mutual potentiation of PTI and ETI leads to effective defense (modified according to https://plantae.org/not-pti-or-eti-pti-and-eti-nature (assessed on 4 February 2025)). NO may induce (arrow) or hinder (stop) signaling pathways, ROS production, or callose deposition differentially if produced early (red) or later (orange) following recognition (for details, see Section 6). HDA, histone deacetylase; MAP(KK)Ks, mitogen-activated protein (kinase kinase) kinases; RBOH, respiratory burst oxidase homolog; RLCK, receptor-like cytoplasmic kinase; ROS, reactive oxygen species; TF, transcription factor.

## Data Availability

Data sharing is not applicable.

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
