# Peer review of "Progress in Plant Nitric Oxide Studies: Implications for Phytopathology and Plant Protection"

_ijms, 2025, doi:10.3390/ijms26052087_

Round 1
Reviewer 1 Report
Comments and Suggestions for Authors
The manuscript "Progress in Plant Nitric Oxide Studies: Implication to Phytopathology and Plant Protection" provides a comprehensive review of the role of nitric oxide (NO) in plant-pathogen interactions and its potential applications in crop protection. It integrates insights from molecular biology, biochemistry, and nanotechnology, bridging fundamental NO biology with applications in sustainable agriculture.
The sections are logically arranged, transitioning smoothly from fundamental NO biology to its applied aspects in plant protection. The section on nanotechnology is promising but lacks sufficient critical analysis of potential risks (e.g., ecological impacts, long-term effects on the soil microbiome). Reformat and reorganize the paragraphs of the last section “Conclusions and future directions”; and addition of emerging technologies (e.g., CRISPR for studying NO pathways or advanced imaging for monitoring NO dynamics) would highlight a forward-looking perspective.
A summary table comparing different NO donors and their efficacy in various experimental systems would be helpful. Although Figure 1 is informative, additional schematic diagrams (or a new figure) summarizing NO's role in specific plant-pathogen interactions or its integration with other signaling molecules (e.g., ROS, SA, and JA) could enhance visual understanding.
The review discusses complex mechanisms of NO synthesis, signaling, and degradation in a detailed and structured manner. It does not extensively discuss the mechanisms underlying NO perception and receptor identification. Current progress suggests that, unlike well-characterized receptor systems for phytohormones, a definitive NO receptor in plants remains elusive. It may be beneficial for the authors to address the absence of a canonical NO receptor in plants and its implications in mediating NO signaling during plant-microbe interactions.
Author Response
Reviewer 1
The manuscript "Progress in Plant Nitric Oxide Studies: Implication to Phytopathology and Plant Protection" provides a comprehensive review of the role of nitric oxide (NO) in plant-pathogen interactions and its potential applications in crop protection. It integrates insights from molecular biology, biochemistry, and nanotechnology, bridging fundamental NO biology with applications in sustainable agriculture. The sections are logically arranged, transitioning smoothly from fundamental NO biology to its applied aspects in plant protection.
Reply: We thank reviewer 1 for valuable comments on our manuscript.
The section on nanotechnology is promising but lacks sufficient critical analysis of potential risks (e.g., ecological impacts, long-term effects on the soil microbiome). Reformat and reorganize the paragraphs of the last section “Conclusions and future directions”; and addition of emerging technologies (e.g., CRISPR for studying NO pathways or advanced imaging for monitoring NO dynamics) would highlight a forward-looking perspective.
Reply: As suggested we have reorganized and improved the last section, including highlights on future perspectives.
A summary table comparing different NO donors and their efficacy in various experimental systems would be helpful.
Reply: We appreciate the reviewer's comment, but we believe this information has been repeatedly presented in previous review papers focused specifically on the molecular details of NO chemistry and chemical biology in biological tissues and cells.
Although Figure 1 is informative, additional schematic diagrams (or a new figure) summarizing NO's role in specific plant-pathogen interactions or its integration with other signaling molecules (e.g., ROS, SA, and JA) could enhance visual understanding.
Reply: We thank the reviewer for his/her suggestions. Figure 1 was modified accordingly to show products of protein PTMs by formula. Moreover, we have added new Figure 2 to provide a more complex overview of NO role in plant defence. Regarding the suggested figure on NO integration with other signalling molecules/phytohormones, we feel this is outside of the focus of our review, while there are excellent recent papers which cover this topic.
The review discusses complex mechanisms of NO synthesis, signaling, and degradation in a detailed and structured manner. It does not extensively discuss the mechanisms underlying NO perception and receptor identification. Current progress suggests that, unlike well-characterized receptor systems for phytohormones, a definitive NO receptor in plants remains elusive. It may be beneficial for the authors to address the absence of a canonical NO receptor in plants and its implications in mediating NO signaling during plant-microbe interactions.
Reply: As suggested, we have added a new paragraph dedicated to the no-existence of the canonical NO-GC-cGCMP signalling pathway operating in animal cells and to accentuate that the NO signalling is not based on the ligand-receptor interaction, but similar to ROS and H2S, NO signalling is based on redox modifications of multiple target proteins.
Reviewer 2 Report
Comments and Suggestions for Authors
The work on this unappreciated cousin of ROS is timely and needed. The section on products of nanosize that could impact RNS plant function is also timely. The fact that the plants' function also involves its microbiome also is a strong point.
The paper does require editing to improve reading- format, choice of words, and sentence structure all are problems that should be addressed to provide a quality product
I also ask for more composite diagrams. The discussions on the mechanisms for defense against pathogens are difficult to assimilate without a diagramatic aid.
They are complex and unfortunately involve so many abbreviations which is OK when you are in the field but a problem if this paper is introductory or of general interest.
Comments are made as sticky notes at the pertinent places.

A review on this topic is timely. But the paper requires editing to improve its quality.
Author Response
Reviewer 2
The work on this unappreciated cousin of ROS is timely and needed. The section on products of nanosize that could impact RNS plant function is also timely. The fact that the plants' function also involves its microbiome also is a strong point. The paper does require editing to improve reading - format, choice of words, and sentence structure all are problems that should be addressed to provide a quality product.
Reply: We thank the reviewer for the critical comments made in the manuscript text. We adopted the majority of the recommended alterations; the recent version reflects the suggestions and comments of all three reviewers. English editing by a native speaker was employed and several parts of the text were rewritten.
I also ask for more composite diagrams. The discussions on the mechanisms for defense against pathogens are difficult to assimilate without a diagramatic aid. They are complex and unfortunately involve so many abbreviations which is OK when you are in the field but a problem if this paper is introductory or of general interest.
Reply: As suggested, Figure 1 was modified to show PTM products by formula as required by the reviewer. New Figure 2 has been added to provide a more complex overview of NO role in the plant defence, esp. to illustrate the complexity of PTI and ETI orchestration
Comments are made as sticky notes at the pertinent places.
Reply:
- Remark on thioredoxin reductase “can you show reaction - what is product?”: in the cited paper by Jedelská et al. (2020), all detailed information is compiled. We, therefore, believe it was not suitable to expand this part of the text.
- Concerning the reviewer´s remarks on ion charge: we thoroughly checked the formatting in the whole text again – both subscripts as well as superscripts were correct. We are sorry, but it is the font used by the IJMS Word template, which may give the impression it is not written properly.
- Remark in Chapter 9 “But I am now confused on the relationship between de novo synthesis and activity due to storage products - can you clarify this issue?” NO metabolism is complex, and the “nitrosative door” to combat pathogens might be influenced by external application, the most easily in post-harvest storage of plant commodities. However, we are aware that elevated NO concentration might lead to PTMs and changes in compositions of proteins, fatty acids, and other nutrients together with the limitation of non-pathogenic microorganisms might possibly influence consumers. However, mankind should look for alternatives to chemical pesticides.
Reviewer 3 Report
Comments and Suggestions for Authors
The review is very well written and compiles in detail the present understanding of nitric oxide (NO) production and its mechanisms of action. The authors put emphasis on NO's role in plant immunity reactions and have properly highlighted the new developments and restrictions in the use of NP-based NO-donors.
My remarks are mainly on some missing visualization or comprehensive arrangement of the information related to chapters 3 and 7:
- I recommend including a scheme (or a table) that visualizes (or lists) the major references on NO-mediated post-translational protein modification.
- The place of NO in the generation of plant immunity responses is not sufficiently understood, as the authors have stated in the text. I suggest including a diagram depicting the general concepts of the plant immunity reactions and the probable positioning of NO in the mechanisms related to the responses (similarly to Figure 1, which gives an excellent visual). Alternatively, the authors may choose to include a table instead, listing briefly the known facts on the participation of NO in plant immunity responses. It could present in brief the case studies, obtained results, and suggested mechanisms of action, as well as the reference.
Some minor mistakes in the text were also spotted (for example on page 6, second line: “….(NO2-Ln) and, for the first time in Arabidopsis, nitro-oleic acid and nitro-linoleic acid.” – there is a missing verb).
The references in the text are given with the names of the authors and alphabetically listed at the end of the text. Usually MDPI style requires for these to be represented by numbers.
Author Response
Reviewer 3
The review is very well written and compiles in detail the present understanding of nitric oxide (NO) production and its mechanisms of action. The authors put emphasis on NO's role in plant immunity reactions and have properly highlighted the new developments and restrictions in the use of NP-based NO-donors.
Reply: We thank the reviewer for his/her positive evaluation of our manuscript.
My remarks are mainly on some missing visualization or comprehensive arrangement of the information related to chapters 3 and 7:
- I recommend including a scheme (or a table) that visualizes (or lists) the major references on NO-mediated post-translational protein modification.
Reply: We appreciate the reviewer's suggestion: However, we this kind of Table has been recently included in a review paper focused on the roles of S-nitrosylation in plants by Borrowman et al., 2023 (Table 1), which is repeatedly cited in our manuscript.
- The place of NO in the generation of plant immunity responses is not sufficiently understood, as the authors have stated in the text. I suggest including a diagram depicting the general concepts of the plant immunity reactions and the probable positioning of NO in the mechanisms related to the responses (similarly to Figure 1, which gives an excellent visual). Alternatively, the authors may choose to include a table instead, listing briefly the known facts on the participation of NO in plant immunity responses. It could present in brief the case studies, obtained results, and suggested mechanisms of action, as well as the reference.
Reply: Figure 1 was modified to show PTM products by formula as required by reviewer 2. New Figure 2 has been added to provide a more complex overview of NO role in the plant defence, esp. to illustrate the complexity of PTI and ETI orchestration
Some minor mistakes in the text were also spotted (for example on page 6, second line: “….(NO2-Ln) and, for the first time in Arabidopsis, nitro-oleic acid and nitro-linoleic acid.” – there is a missing verb).
Reply: We have thoroughly checked the entire manuscript to correct these and other mistakes.
The references in the text are given with the names of the authors and alphabetically listed at the end of the text. Usually MDPI style requires for these to be represented by numbers.
Reply: We changed formatting of citations according to the IJMS Guide to Authors.
Round 2
Reviewer 1 Report
Comments and Suggestions for Authors
The manuscript has been sufficiently improved to warrant publication in IJMS.
Author Response
We thank all reviewers for their valuable comments
Reviewer 2 Report
Comments and Suggestions for Authors
thanks for your efforts to update this important and highly interactive research area
the emphasis on NO production by plant-associated organisms is especially needed

there are sections in the reworked portions that are unclear
suggestions are made as sticky notes
Author Response
We thank all reviewers for their valuable help. All comments of reviewer 2 were taken into consideration, except for p. 5 - thiyl radical – it is correct in the present form, please see papers e.g. https://www.sciencedirect.com/science/article/pii/S0021925819302510 or https://pmc.ncbi.nlm.nih.gov/articles/PMC5118943/ , etc.